# A Minimally-Invasive, Simple, Rapid, and Effective Surgical Technique for the Treatment of Ingrown Toenails: A Reminder of the Original Winograd Procedure

**DOI:** 10.3390/ijerph18010278

**Published:** 2021-01-01

**Authors:** Jahyung Kim, Sanghyeon Lee, Jeong Seok Lee, Sung Hun Won, Dong Il Chun, Young Yi, Jaeho Cho

**Affiliations:** 1Department of Orthopedic Surgery, Seoul Hospital, Soonchunhyang University Seoul Hospital, Seoul 04401, Korea; hpsyndrome@naver.com (J.K.); 124856@schmc.ac.kr (J.S.L.); orthowon@gmail.com (S.H.W.); orthochun@gmail.com (D.I.C.); 2Department of Orthopaedic Surgery, Ewha Womans University Seoul Hospital, Seoul 07804, Korea; snnov9@gmail.com; 3Department of Orthopedic Surgery, Seoul Paik Hospital, Inje University College of Medicine, Seoul 04551, Korea; 20vvin@naver.com; 4Department of Orthopaedic Surgery, Chuncheon Sacred Heart Hospital, Hallym University, Gangwon-do 24253, Korea

**Keywords:** ingrown toenail, matrixectomy, minimal-invasive

## Abstract

(1) Background: Ingrown toenail is a common disorder of the toe that induces severe toe pain and limits daily activities. The Winograd method, the most widely used operative modality for ingrown toenails, has been modified over years to include wedge resection of the nail fold and complete ablation of the germinal matrix. We evaluated the outcomes of original Winograd procedure without wedge resection with electrocautery-aided matrixectomy. (2) Methods: We retrospectively analyzed the outcomes of patients who underwent surgery for ingrown toenails at a university hospital for two years from November 2015 to October 2017. Surgery was performed in 76 feet with a mean operation time of 9.34 min. (3) Results: The minimal interval from surgery to return to regular activities was 13.26 (range 7 to 22) days. Recurrence and postoperative wound infections were found in 3 (3.95%) and 2 (2.63%) patients, respectively. Evaluation of patient satisfaction at one-year follow-up showed that 40 (52.63%) patients were very satisfied, 33 (43.42%) were satisfied, 3 (3.95%) were dissatisfied, and none of them were very dissatisfied. The average follow-up duration was 14.66 (range 12 to 25) months. (4) Conclusions: Therefore, it is believed that this less-invasive and simple procedure could be easily performed by clinicians, with satisfactory patient outcomes.

## 1. Introduction

Ingrown toenail, also called onychocryptosis, is a common disorder involving big toes. Although considered a relatively minor foot ailment, ingrown toenail causes severe toe pain when walking, and eventually limits routine daily activities [1,2]. Besides, if a patient has diabetes mellitus, ingrown toenail can deteriorate into a diabetic foot ulcer and induce chronic osteomyelitis of the involved limb [3]. Therefore, adequate management of ingrown nails is crucial.

In the early stages, ingrown toenails can be managed successfully via nonsurgical methods including foot care (avoiding ill-fitting footwear and soaking in warm water), topical oral antibiotics, proper nail trimming and elevation of nail corner. However, operative intervention is required if conservative treatment fails or the toenail is accompanied by seropurulent drainage or chronic inflammation [4,5].

Among the various surgical treatments for ingrown toenail, the Winograd method is a popular technique that has been used frequently since 1927, and entails partial nail plate excision and curettage of nail bed and germinal matrix [1]. In an effort to avoid recurrence, such minimally invasive procedures have been modified over the years to include wedge resection of the nail fold and complete ablation of germinal matrix via chemical or electrical matrixectomy [1,6,7,8,9]. However, based on a review of previous studies, poor postoperative outcome related to delayed wound healing and recurrence rate have been detected in wedge resection procedures [6].

Therefore, we postulated that wedge resection of the germinal fold, a relatively invasive procedure, could be more harmful than beneficial. The original Winograd method for partial excision of only the nail plate is of interest.

In the present study, we combined the original Winograd procedure with matrixectomy using electrocoagulation in patients with ingrown toenail. We retrospectively reviewed the data showing the recurrence rate, post-operative infection rate, surgical duration, the time required to return to regular activities, and patient satisfaction.

## 2. Materials and Methods

We retrospectively analyzed and compared the outcomes of patients. The purpose of this study was to evaluate the patients who underwent surgery for ingrown toenail at a university hospital for two years from November 2015 to October 2017. On 29 April 2020, this study was approved by the Ethics Committee of Chuncheon Sacred Heart Hospital, Hallym University (Institutional Review Board number: CHUNCHEON 2020-04-006).

### 2.1. Design and Sample

Sixty-nine patients underwent surgical treatment for ingrown toenail by one surgeon. The patients included 7 cases with ingrown toenails on both feet and each foot was counted separately. Patients who underwent previous surgery for ingrown toenail were excluded. A total of 76 great toes were treated surgically. Preoperative Heifetz staging revealed 68 cases of stage 2 and 8 cases of stage 3. The patient was referred to the hospital for dressing change the day after surgery, and sutures were removed about 2 weeks after surgery. A follow-up phone call 1 year after the surgery was conducted to determine the recurrence and the degree of satisfaction. Patient satisfaction was categorized as follows: very satisfied, satisfied, dissatisfied, and very dissatisfied [7]. We recorded the operative time, the time required to return to regular activities and the incidence of postoperative infection. We defined recurrence as evidence of (1) ingrown nail edge, (2) spicule formation from the germinal matrix, and (3) recurrence of previous symptoms [8].

### 2.2. Operative Method

Surgical treatment was performed by a single surgeon. Thirty minutes before the operation, 1 g of cefazolin was injected intravenously, and after the surgery, the first-generation cephalosporin was administered orally. Adjunctive antibiotic use was limited to the patients with definite infection signs including substantial erythema or purulent drainage. The patient was placed on the operating table. To induce local anesthesia, a digital nerve block was performed at the base of the affected big toe using 1% lidocaine solution. The rubber band tourniquet was applied at the base of the toe. A small incision (less than 10 mm) was made at the eponychium along the line of expected nail excision, followed by a gentle, blunt dissection with the Freer elevator to separate the soft tissue from the ingrown portion of the nail. After the dissection, a partial excision (a quarter to one-fifth) nail plate was performed with a small scissor. Using a surgical curette, the germinal matrix and nail bed were scraped off and destroyed. The granulation tissue at the lateral margin of the nail plate was incised using a scalpel without resection of the lateral nail fold. The germinal matrix and nail bed were destroyed via electrocautery in coagulation mode for a total of 3 to 4 s. After copious irrigation, one nylon suture was applied to attach the resected eponychium. In case of unstable nail plate following the gentle, blunt dissection of the soft tissue, additional distal suture was made to stabilize the nail plate. (Figure 1). 

## 3. Results

The 76 cases included 40 (52.63%) males and 36 (47.37%) females with a mean age of 41.18 years (range 12 to 85). The mean operation time was 9.34 min, without any surgical complications.

The minimal interval from surgery to return to regular activities was 13.26 (range 7 to 22) days. The follow-up of patients at one year revealed that 40 (52.63%) were very satisfied, 33 (43.42%) were satisfied, 3 (3.95%) were dissatisfied, and none were very dissatisfied. The altered rate of satisfaction was attributed to postsurgical pain due to residual scar. Recurrence was observed in 3 (3.95%) patients and postoperative wound infection was found in 2 (2.63%) cases. Lastly, the average follow-up duration was 14.66 (range 12 to 25) months (Table 1).

## 4. Discussion

Unless the ingrown nail is limited to early inflammation, operative treatment is indicated in many cases. Among a variety of surgical methods, the ideal procedure should result in a low recurrence rate, have a short interval of return to regular activity, and be easy to perform [5,9,10]. In the current study, recurrence and infection rates were relatively low at 3.95% and 2.63%, respectively. Further, the time to return to regular activity was short, and the surgical technique was relatively easy as no additional nail fold resection was performed.

The symptoms of ingrown toenail manifest when the paronychium is in contact with the lateral tip spicule of the nail plate, which causes irritation and reaction to a foreign body that eventually lead to inflammation [10,11]. In an effort to avoid skin irritation by the nail plate, special suture techniques were used to lay the skin under the nail after wedge resection, which actually reduced the recurrence rate [10,11]. Furthermore, in order to radically remove the inflammatory granulation tissue formed by repetitive irritation of the lateral fold, wedge resection of the nail fold has been widely reported in the literature, invariably followed by complete skin closure [8,12].

However, as the resected skin edges of the lateral fold are likely to be attached to the nail rather than the skin, the suture is improperly approximated. One of the many principles of wound closure is that the bound skin edges must be aligned well such that there is no difference in height or gap between the edges. Improperly bound skin edges caused by incorrect wound closure induce the formation of granulation tissue and wound contraction [13]. Besides, underneath the skin margin of the approximated wedge resection, the sterile matrix layer below the nail and resected lateral fold soft tissue are improperly aligned and form the raw surface. Thus, it not only delays the healing time by increasing pain, discharge and infection rates, but also induces granulation tissue formation and contracture, which contribute to recurrent symptoms. Recently, the technique of suturing the subcutaneous tissue and skin by introducing a subungual suture after wedge resection has been reported [14]. This technique is not easy, but it is considered to have the advantage of expecting good cosmetic results without nail bed injury.

In our cases, we did not perform wedge resection for the lateral nail fold after the excision of partial nail plate, leaving the space next to the lateral fold empty. In this minimally invasive procedure, not only did the remaining nail plate not irritate the skin, but also the lateral fold prevented delayed complications associated with wound healing such as scar pain and infection. It also resulted in concomitant cosmetic effects due to the absence of scar formation. These advantages contribute to superior patient satisfaction and a relatively short time required for the return to regular activity compared with previous studies associated with wedge resection [5,7,10,15]. Although wedge excision of the nail fold together with toenail, germinal matrix, and nail bed is commonly considered an inevitable component of Winograd method in the literature, the original procedure first suggested by Winograd in 1929 was less invasive, excluding wedge excision [12]. Accordingly, we agree with Winograd that partial toenail ablation without supplementary nail fold excision is an effective procedure that can be easily performed without substantial learning curves. 

However, the original Winograd technique was modified due to symptom recurrence including pain, swelling, redness, and discharge. Such symptoms are caused by re-growth of nail plate or invasion of distally angled skin above the nail plate [6,10,16]. As the germinal matrix is the source of the bulk of the nail plate, improperly destroyed matrix may induce regrowth of the nail plate at the distally angled skin, causing recurrent symptoms [7,8,9,15]. In order to prevent re-growth of the nail plate, a variety of methods have been used to destroy the germinal matrix via surgical curettage in addition to chemical matrixectomy or electrocauterization [4,7,8]. Although the original Winograd procedure also entails matrixectomy, it is solely performed via surgical curettage and may not be enough to radically destroy the productive center of nail plate, resulting in a relatively high recurrence rate of the ingrown toenail [8]. The recurrence rate with surgical curettage combined with wedge resection was reported to range from 7.7% to 17.7%, which might have been related to limited dissection and destruction of the lateral horn of the germinal matrix from the underlying periosteum [5,6,8].

Therefore, additional treatment is required to ensure complete destruction of the lateral horn of the germinal matrix. Chemical matrixectomy can be easily accomplished using a chemical solution at the surgical site without special surgical techniques. It is known to be effective in preventing recurrence and regrowth of the ingrown toenail than electrocauterization and is commonly indicated for Heifetz stage 2 and 3 ingrown toenails [17]. However, needless application of chemical agents to the tissue surrounding the germinal matrix can lead to unpredictable tissue damage and chemical burn, which may lead to infection, increased drainage, and prolong wound healing time [4,8,9]. Although Alvarez-Jimenez et al. reported significantly reduced mean wound healing time and infection rate with the use of segmental phenolization through the randomized controlled study, it may also lead to greater postsurgical bleeding and pain [18].

On the other hand, if matrixectomy is performed by electrocauterization, the germinal matrix can be destroyed precisely at the target with decreased risk of damage to the surrounding tissue. Such accurate targeting is facilitated via small incisions to the eponychium for better visualization of the lateral margin of germinal matrix as in our case. Acar et al. reported no recurrence in patients who had undergone surgery with the Winograd procedure combined with wedge resection, electrocoagulation, and traditional sutures while a recurrence rate of 6% was reported in the group without electrocoagulation [7]. The recurrence rate of our procedure that supplemented the original Winograd method with electrocautery-aided matrixectomy was compatible with these results. Combined with a lower rate of recurrence and the short time for return to regular activity, up to 96% of the patients were satisfied after the surgery, which was higher compared with previous studies using wedge resection [7,10].

Because ingrown toenails are prone to infections commonly caused by species like *Staphylococcus aureus*, surgeons have used oral antibiotics before or after surgery without established clinical evidence [4]. Nevertheless, studies have detected no significant difference in clinical outcome between a group that received antibiotics along with surgery and a group that solely underwent surgery [19,20]. These studies advocate that local infection eventually resolves after the nail excision and matricectomy. In line with such findings, satisfactory clinical outcome could be achieved in this study even though an adjunctive antibiotic use was merely limited to the patients with definite infection signs. 

The average operation time in this study was 9.34 min. In the previous studies, although not directly compared with our results, higher operation times were reported by authors who performed wedge resection, which varied from 13 to 15.7 min [5,16,21]. Such differences might be attributed to the simplified surgical procedure along with exclusion of nail fold wedge resection and the absence of complex suturing process. In addition, the destruction of germinal matrix using electrocautery required a total of only 3 to 4 s, which might have contributed to a reduced operation time. This simplified, minimally invasive procedure can be easily performed by all clinicians without a steep learning curve.

This study has some limitations. First, it is a retrospective study, with no control group for comparison. Thus, randomized controlled trials or prospective studies are needed to compare the outcomes of minimally invasive procedures with relatively invasive procedures including wedge resection. In addition, electrocauterization would have to be compared with chemical matricectomy in the succeeding study in order to prove its effectiveness in destroying the germinal matrix. Second, it cannot be said that the surgical procedure suggested in this study can be applied to all kinds of ingrown toenails. For example, nail extraction would be favored in the case of thickened or deformed toenail while a nail implant would rather be used in case of narrow toenail with incurvatum pattern. In other words, different therapeutic method would have to be chosen upon the shape or width of the patient’s toenail. Nevertheless, we propose that our surgical procedure would be valuable for general cases because matrixectomy is the most commonly used procedure for typical ingrown toenail. Third, although this study included ingrown toenails of both Heifetz stage 2 and 3, only eight cases (10.53%) were in stage 3. Since the original Winograd procedure is especially indicated in stages of onychocryptosis with the presence of granulation tissue (Heifetz stage 3), the result may have been influenced, referring to the low rate of postoperative infections. Lastly, because full width of the toenail becomes narrowed after partial excision of the nail plate, patients who put priority on aesthetic issue may be less satisfied with the proposed surgical technique.

## 5. Conclusions

A simplified, rapid, and less invasive surgical procedure is ideal for the treatment of ingrown toenail. In addition, complete matrixectomy is needed in order to diminish the recurrence rate. Taken together, the original Winograd procedure (without wedge resection) combined with electrocauterization of germinal matrix is a minimally invasive, simple, rapid, and effective treatment modality for ingrown toenails, resulting in superior patient satisfaction.

## Figures and Tables

**Figure 1 ijerph-18-00278-f001:**
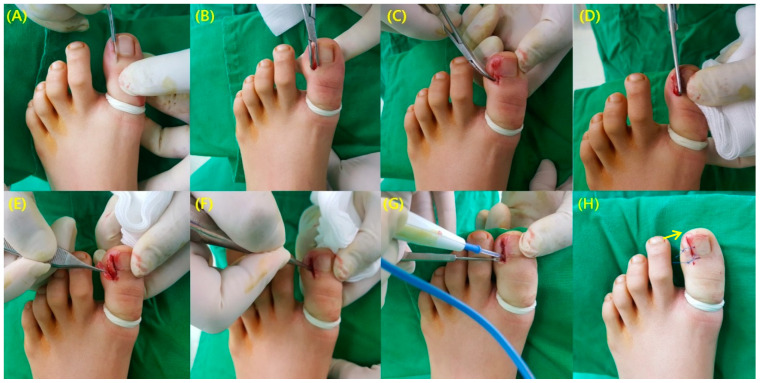
Sequences of operative procedure (**A**) Less than 10mm incision made at the eponychium. (**B**) Blunt dissection to separate the soft tissue from the ingrown part of the nail. (**C**) Exposure of the overgrown nail plate. (**D**) Partial excision of the nail plate. (**E**) Excised portion of the nail plate (1/4 to 1/5 of the nail). (**F**) Curettage of the germinal matrix and nail bed. (**G**) Additional destruction of germinal matrix using electrocautery. (**H**) One nylon suture was performed to attach the resected eponychium. In case of unstable nail plate following the gentle, blunt dissection of the soft tissue, additional distal suture was made to stabilize the nail plate (arrow).

**Table 1 ijerph-18-00278-t001:** Clinical patient characteristics and outcome after surgery.

	Total (N = 76)
Age (year)	41.18 ± 21.73 (12 to 85)
Sex	
Female	36 (47.37%)
Male	40 (52.63%)
Heifetz Stage	
2	68 (89.47%)
3	8 (10.53%)
Operation Time (min)	9.34 ± 3.59 (5 to 15)
Satisfaction (n(%))	
Very satisfied	40 (52.63%)
Satisfied	33 (43.42%)
Dissatisfied	3 (3.95%)
Very dissatisfied	0(0%)
Recurrence (n(%))	
No	73 (96.05%)
Yes	3 (3.95%)
Postop Infection (n(%))	
No	74 (97.37%)
Yes	2 (2.63%)
Follow up period (Month)	14.66 ± 3.64 (12 to 25)
Interval to return to regular activities (days)	13.26 ± 1.98 (7 to 22)

## Data Availability

The data presented in this study are available on request of the authors.

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
