# Peer review of "A Minimally-Invasive, Simple, Rapid, and Effective Surgical Technique for the Treatment of Ingrown Toenails: A Reminder of the Original Winograd Procedure"

_ijerph, 2021, doi:10.3390/ijerph18010278_

Round 1

Reviewer 1 Report

Wedge resection is indeed associated with high recurrence and their use should be limited. In contrast, partial chemical matricectomy treatments have become the gold standard, especially in the early lesions of ingrown nails.

Generally, the principle of ingrown toenail surgery is based on choosing the procedure for the patient. One approach should not be used for all patients.

If the periungual shaft hypertrophy is dominant - remove it. If the problem of the nail is dominant, which is additionally screwed in, the method of narrowing the nail is selected.

76 big toes is a sufficient number of cases. The group indicates 2nd degree dominance (less advanced changes). The satisfaction rating is very general. It could be assessed in more detail which aspects of appearance, pain, or recovery from surgery are considered satisfactory by the patient. It also gives the surgeon room for verification in specific areas.

Controversial according to WHO is the use of antibiotic therapy for the above-mentioned treatments.
The use of suture through the nail plate is damaging to the nail bed - they should be avoided. In addition, 1 seam is enough. The sutetre does not matter here - if it is oversized and scarred, it should be cut, not suture. If it's not overgrown, healing should be just as good without suturing.

The study should compare partial chemical matricectomy versus electrocoagulation
The two inflammatory complications are not described as what they involved. It is quite a high percentage for such a minimally invasive procedure. This may indicate an incorrect selection of a patient, e.g. with a high inflammation of the shafts, in which the shafts should be removed. Recurrence is at a low level comparable to chemical partial matrixectomy methods. But you should compare these methods directly.

In fact, the technique is easy to learn, but in some countries the aesthetics, especially the full width of the nail, is very important. Any technique of narrowing the nail may be less acceptable, especially in the female group.
Surgical curettage combined with ranged wedge resection should be prohibited
Chemical matricectomy can be easily performed using a chemical solution at the site of surgery without special surgical techniques. Therefore, the work had to compare both methods.

Author Response

Wedge resection is indeed associated with high recurrence and their use should be limited. In contrast, partial chemical matricectomy treatments have become the gold standard, especially in the early lesions of ingrown nails.

à Thank you for your comment. We totally agree with your opinion on the high recurrence rate of wedge resection and provided an explanation for its possible complications on the discussion (Line 137-146). Although partial chemical matricectomy is considered to be useful in the early lesions of ingrown nails, it has been reported that it may bring out inevitable application of chemical agents to the tissue surrounding the germinal matrix, followed by unpredictable tissue damage and chemical burn. In an effort to overcome such drawbacks that may lead to infection, increased drainage, and prolonged, we chose electrocauterization. We described a detailed explanation about it on the discussion (Line 175-192).

Generally, the principle of ingrown toenail surgery is based on choosing the procedure for the patient. One approach should not be used for all patients.

If the periungual shaft hypertrophy is dominant - remove it. If the problem of the nail is dominant, which is additionally screwed in, the method of narrowing the nail is selected.

à Thank you for your detailed advice toward our work. We fully agree on your opinion that different therapeutic method should be chosen upon respective individuals. While nail extraction would be favored in case of thickened or deformed toenail, nail implant would rather be used in case of narrow toenail with incurvatum pattern. Instead, we report surgical outcome of matrixectomy used for common type of ingrown toenails with the use of a minimally invasive, simple, rapid, and effective surgical technique. We added such explanation as the limitation of our study (Line 220-227).

76 big toes is a sufficient number of cases. The group indicates 2nd degree dominance (less advanced changes). The satisfaction rating is very general. It could be assessed in more detail which aspects of appearance, pain, or recovery from surgery are considered satisfactory by the patient. It also gives the surgeon room for verification in specific areas.

àThank you for your considerate comment.

Controversial according to WHO is the use of antibiotic therapy for the above-mentioned treatments.

à Thank you for your comment. Patients included in this study were accompanied with relatively moderate to severe lesions of Heifetz stage 2 or 3. Apart from severe, disabling pain, adjunctive antibiotics were used before the surgery in the presence of definite infection sign like substantial erythema or purulent drainage. We added such therapeutic option on the material and method section. However, controversy exists in the literature in terms of adjunctive antibiotics use in ingrown toenail. Therefore, we summarized the use of adjunctive antibiotics in ingrown toenail (Line 198-205).

The use of suture through the nail plate is damaging to the nail bed - they should be avoided. In addition, 1 seam is enough. The sutetre does not matter here - if it is oversized and scarred, it should be cut, not suture. If it's not overgrown, healing should be just as good without suturing.

à Thank you for your comment. We agree with your opinion that the use of suture through the nail plate should be avoided because it can damage the nail bed and one seam would be enough in inevitable cases. Because small incision was made on the eponychium, only one skin suture was made to approximate the incised eponychium, without invading the nail bed. Instead, additional suture was performed only in the case unstable nail plate following the gentle, blunt dissection of the freer. We added such detailed condition for additional suture on the manuscript and Figure 1 (Line 97-100).

The study should compare partial chemical matricectomy versus electrocoagulation

The two inflammatory complications are not described as what they involved. It is quite a high percentage for such a minimally invasive procedure. This may indicate an incorrect selection of a patient, e.g. with a high inflammation of the shafts, in which the shafts should be removed. Recurrence is at a low level comparable to chemical partial matrixectomy methods. But you should compare these methods directly.

à Thank you for your comment. We definitely agree with your opinion that the most common procedure to treat locally infected ingrown toenail is partial avulsion of the lateral edge of the nail followed by chemical matricectomy using 80 to 88% phenol (Heidelbaugh, Management of the Ingrown Toenail, American Family Physician Volume 79, Number 4 â—† February 15, 2009). Nevertheless, studies on electrocauterization as a way to destroy germinal matrix have been reported recently. In addition, in an effort to avoid surrounding tissue damage arisen by chemical solution, we chose electrocauterization which can effectively destroy germinal matrix without expertise surgical technique in minimum time of three to four seconds. We also are in line with your opinion that comparative study between chemical matricectomy and electrocauterization would be needed. We added it as a limitation of the study and provided detailed explanation about two procedures in the discussion (Line 175-192, 218-220).

In fact, the technique is easy to learn, but in some countries the aesthetics, especially the full width of the nail, is very important. Any technique of narrowing the nail may be less acceptable, especially in the female group.

àThank you for your comment and we agree with your opinion that cosmetic issue was not considered in this technique. We added such explanation as the limitation of this study (Line 231-233).

Surgical curettage combined with ranged wedge resection should be prohibited

Chemical matricectomy can be easily performed using a chemical solution at the site of surgery without special surgical techniques. Therefore, the work had to compare both methods.

à Thank you for your comment. As we mentioned above, we definitely agree with your opinion.  We added it as a limitation of the study and provided detailed explanation about two procedures in the discussion (Line 218-220).

Reviewer 2 Report

Within this manuscript the authors describe an interesting restrocpective study of original Winograd procedure.

Some matters which are important for the conclusion are not clearly described in the manuscript.

The original Winograd procedure is especially indicated in stages of onychocryptosis with the presence of granulation tissue (Heifetz stage 3). In the Design and Sample section it is specified “A total of 76 great toes were treated surgically with Preoperative Heifetz staging”. This revealed 68 cases of stage 2” and only 8 cases (10.53%) of stage 3. We consider the need to include this fact as a limitation of the study in the discussion section, since that this circumstance may have influenced the results, especially referring to the low rate of reported post-surgical infections.

The authors report lower rate of recurrence. However, in the discussion section it is specified that the original Winograd technique was modified due to symptom recurrence. The current available evidence shows that “the addition of phenol is probably more efective in preventing recurrence and regrowth of the ingrowing toenail” and Chemical matrixectomy is indicated in both stage 2 and 3 of Heifetz.  (Interventions for ingrowing toenails. Cochrane Database Syst Rev.)

While is true that the application of chemical agents to the tissue surrounding the germinal matrix can lead to unpredictable tissue damage, which may lead to infection, increased drainage, and prolong wound healing time. However, in discussion section it should be included that these drawbacks have been reduced with modification of the phenol technique that has shown their effectiveness in clinical trials.  We recommend including the following reference:

Álvarez-Jiménez J, Córdoba-Fernández A, Munuera PV. Effect of curettage after segmental phenolization in the treatment of onychocryptosis: a randomized double-blind clinical trial. Dermatol Surg 2012, 38,454-461.

Minor comments

References 15 and 17 must be corrected (journal is missing).

Author Response

Within this manuscript the authors describe an interesting restrocpective study of original Winograd procedure.

àThank you and we appreciate your positive perspective toward our work.

Some matters which are important for the conclusion are not clearly described in the manuscript.

The original Winograd procedure is especially indicated in stages of onychocryptosis with the presence of granulation tissue (Heifetz stage 3). In the Design and Sample section it is specified “A total of 76 great toes were treated surgically with Preoperative Heifetz staging”. This revealed 68 cases of stage 2” and only 8 cases (10.53%) of stage 3. We consider the need to include this fact as a limitation of the study in the discussion section, since that this circumstance may have influenced the results, especially referring to the low rate of reported post-surgical infections.

à Thank you for your comment and included the fact you mentioned on the limitation (Line 227-231).

The authors report lower rate of recurrence. However, in the discussion section it is specified that the original Winograd technique was modified due to symptom recurrence. The current available evidence shows that “the addition of phenol is probably more efective in preventing recurrence and regrowth of the ingrowing toenail” and Chemical matrixectomy is indicated in both stage 2 and 3 of Heifetz. (Eekhof, Interventions for ingrowing toenails. Cochrane Database Syst Rev., 2012 Apr 18;(4):CD001541) While is true that the application of chemical agents to the tissue surrounding the germinal matrix can lead to unpredictable tissue damage, which may lead to infection, increased drainage, and prolong wound healing time. However, in discussion section it should be included that these drawbacks have been reduced with modification of the phenol technique that has shown their effectiveness in clinical trials.  We recommend including the following reference:

Álvarez-Jiménez J, Córdoba-Fernández A, Munuera PV. Effect of curettage after segmental phenolization in the treatment of onychocryptosis: a randomized double-blind clinical trial. Dermatol Surg 2012, 38,454-461.

à Thank you for your comment. Original Winogrand technique includes partial nail plate excision and curettage of nail bed and germinal matrix. As we mentioned on the introduction, such minimally invasive procedure has been modified over the years to include wedge resection of the nail fold and complete ablation of germinal matrix using chemical or electrical matrixectomy, in an effort to avoid recurrence. In this study, we reminded of orginal Winograd technique by avoiding wedge resection that may delay wound healing and cause postoperative infection. In addition, we chose electrical matrixectomy in terms of complete ablation of germinal matrix. It is true that addition of phenol is probably more effective in preventing recurrence and regrowth of the ingrown toenail, as mentioned on the systematic review by Eekhof et al (Eekhof, Interventions for ingrowing toenails. Cochrane Database Syst Rev., 2012 Apr 18;(4):CD001541). This study also states that chemical matrixectomy is indicated in both stage 2 and 3 of Heifetz. However, we were not able to use phenol for matrixectomy because 80 to 85% phenol was not available for use in our hospital. As an alternative, we used electrocauterization that is easier to be used.

We added your recommendations in the manuscript with the reference (Álvarez-Jiménez J, Córdoba-Fernández A, Munuera PV. Effect of curettage after segmental phenolization in the treatment of onychocryptosis: a randomized double-blind clinical trial. Dermatol Surg 2012, 38,454-461.) you provided us (Line 176-185).

“Although Alvarez-Jimenez et al reported significantly reduced mean wound healing time and infection rate with the use of segmental phenolization through the randomized con-trolled study, it may also lead to greater postsurgical bleeding and pain”

Thank you again for your considerate advice toward our work.

Minor comments

References 15 and 17 must be corrected (journal is missing).

àThank you and we modified the references.

Round 2

Reviewer 1 Report

86-88 accept

98-100 and 107-09 accept

138 gluteal ford?

138-147 There are techniques in which, after removing the nail shaft, the edge to the edge of the skin is attached with a subungual suture according to the surgical rules. The technique is not easy but effective with few complications. It is worth citing this work in the discussion. https://www.sciencedirect.com/science/article/pii/S2049080120301680

177-186 I agree

187-193 accept

199-206 I agree

Author Response

Reviewer 1 R2

86-88 accept

à Thank you for your accept

98-100 and 107-09 accept

à Thank you for your accept

138 gluteal ford?

à I will clarify the gluteal fold with “the lateral fold.”

138-147 There are techniques in which, after removing the nail shaft, the edge to the edge of the skin is attached with a subungual suture according to the surgical rules. The technique is not easy but effective with few complications. It is worth citing this work in the discussion. https://www.sciencedirect.com/science/article/pii/S2049080120301680

à Thank you again for your considerate advice toward our work.

We added your recommendations in the manuscript with the reference (Dabrowski, M.; Litowinska, A. Recurrence and satisfaction with sutured surgical treatment of an ingrown toenail. Ann Med Surg (Lond) 2020, 56, 152-160, doi:10.1016/j.amsu.2020.06.029.) you provided us (Line 144-147).

“Recently, the technique of suturing the subcutaneous tissue and skin by introducing a subungual suture after wedge resection has been reported. [14] This technique is not easy, but it is considered to have the advantage of expecting good cosmetic results without nail ned injury.”

177-186 I agree

à Thank you for your agreement.

187-193 accept

à Thank you for your accept.

199-206 I agree

à Thank you for your accept

This manuscript is a resubmission of an earlier submission. The following is a list of the peer review reports and author responses from that submission.